# Burial leakage: A human accustomed groundwater contaminant sources and health hazards study near cemeteries in Benin City, Nigeria

Ifeanyi Maxwell Ezenwa[1], Michael Omoigberale[2], Rachel Abulu[2,3], Ekene Biose[4], Benjamin Okpara[2], Osariyekemwen Uyi[2,5,6]*

1 Department of Zoology and Environmental Biology, University of Nigeria, Nsukka, Enugu, Nigeria, 2 Department of Animal and Environmental Biology, Faculty of Life Sciences, University of Benin, Benin City, Nigeria, 3 Geography Department, N431 Rose Building, York University, Toronto, Canada, 4 Department of Environmental Management and Toxicology, University of Benin, Benin City, Nigeria, 5 Department of Zoology and Entomology, Faculty of Natural and Agricultural Sciences, University of the Free State, Bloemfontein, South Africa, 6 Department of Entomology, University of Georgia, Tifton, GA, United States of America

* osariyekemwen.uyi@uniben.edu

**Data Availability Statement:** All relevant data are within the paper and its Supporting Information files.

## Abstract

This study was carried out to assess the levels of physico-chemical parameters that could be impacted by burial leakage and associated human health risks in Benin City, Nigeria. A total of thirty groundwater samples were collected from two cemeteries and analysed for pH, alkalinity, chloride, sulphate, nitrate, phosphate, ammonia- N, calcium, sodium, potassium, $BOD_5$, COD, Mn, Cd, Cu, Ni, Pb, Zn and Fe. The concentrations of the parameters were compared to national and international standards. The results revealed that the groundwater is highly acidic in nature. Principal component analysis (PCA) revealed that except for alkalinity, all other parameters characterised contributed significantly to various principal components (PC) with eigenvalues $\geq$ 1. Moreover, the significance of the PC depicted decomposition of the body corpse and associated burial materials. Water quality index (WQI), heavy metal evaluation index (HEI) and Nemerov pollution index (NI) indicated that groundwater from the study area is of poor quality, and highly contaminated by heavy metals. We determined the Chronic health risk through exposure by calculating the hazard quotient (HQ) and hazard index (HI), for both children and adults. For the oral exposure, approximately 33% of samples suggest the high category of chronic risk for children while the medium category was indicated for adults. We found that oral exposure showed relatively higher risk than dermal exposure, and chronic risk for children and adults ranged from low to negligible. However, the carcinogenic risk of Ni and Pb via oral exposure route suggests, very high risk for Ni and medium risk for Pb. In consideration that long term exposure to low concentrations of some heavy metals (including Pb, Cd, and Ni) could result in different manifestations of cancer, we recommend that residents of these areas should find an alternative source of water for drinking and other domestic uses.

**Funding:** The authors received no specific funding for this work.

**Competing interests:** No: The authors have declared that no competing interests exist.

## Introduction

Many urbanised and rural communities rely heavily on groundwater for agricultural activities drinking water and other purposes [1, 2]. The quality of groundwater is known to be strongly influenced by anthropogenic interference with the hydrological cycle which may manifest as the leaching of nutrients or pesticides under cultivated lands [3–8]. Other human activities that can impact the quality of groundwater include municipal waste disposal processes, natural resource exploration, interment of human remains after death, commercial and industrial activities etc. As a result, contaminated groundwater may lower the quality of drinking water, reduce water supply, increase prices for alternate water sources, and cause some substantial health issues [2, 6].

In some cultural or religion affiliations across the globe, cemetery remains an indispensable component of landscape setting as it stands as the last space to accommodate the remains of human after death. On scientific concept, cemeteries are intended to keep the biogeochemistry cycle going but every system on earth maintains certain carrying capacity or threshold beyond which adverse effect prevails. By interment of human remains after death into the ground, man unintentional contribute to fasten the release of contaminants into the groundwater; a condition which is dependent on local strata and sediment of geology of the area [9]. Although cemeteries offer some ecosystem services, such as storm water infiltration, green space, and micrometeorology control, they also pose a risk as potential sources of groundwater contamination [7]. According to a number of authors, cemeteries serve as specialized forms of landfill due to the interactions between soil and degradable materials (body parts, coffins, etc.) [7]. Inorganic and organic components that have the potential to contaminate various environmental media, including groundwater, may be released due to the deterioration and decomposition of human bodies and casket components over time. Several studies from different parts of the World including Australia, Brazil, Iran, South Africa, Portugal, United States of America and United Kingdom have demonstrated that interment of human remains after death has potential of contaminating groundwater [8–18]. In addition to serving as the final resting place for bodies, cemeteries also house the coffins and caskets used for interment. According to Fineza et al. [10], minerals used to make coffins constitute a major source of contamination found in cemeteries. When these minerals corrode, dangerous toxic substances are released. Thus, improper positioning or selection of cemetery site could pose high level of threat to human health and the environment in general. Jonker and Olivier [19] reported that the chemicals used in coffin-making may diffuse into the surrounding soils via seepage from the graves. From there, they might seep into the groundwater and pose a threat to the health of locals living close to the cemetery. Wood preservatives and paints used in casket construction contain chemicals such as ammoniac or chromated copper arsenate and copper naphthalene [9]. Heavy metals such as cadmium, chromium, lead, mercury and barium are major constituents of the paints. Barium is used as a pigment and a corrosion inhibitor while arsenic serves as a pigment, wood preservative and anti-fouling ingredient [19, 20].

Across the globe, studies conducted at different time scales and environmental conditions have revealed that leakage from cemeteries constitute major source of contaminant to groundwater. Pioneer research in this regard was published in 1951 by Van Haaren [21] who measured high concentrations of some ions including chlorides (500 mg L$^{-1}$), sulphates (300 mg L$^{-1}$), bicarbonates (450 mg L$^{-1}$) and conductivity (2300 μS/cm) in shallow aquifer within the vicinity of a cemetery. Alagbe et al. [22] reported slightly acidic pH and elevated levels of high salinity, and lead in groundwater samples obtained around Ayobo Cemetery in Lagos, Nigeria. Turajo et al. [23] explored the issues of burial practice and its effect on groundwater pollution in Maiduguri, Nigeria and reported that the groundwater samples from boreholes at very close

proximity to the cemetery were characterised by high pH values among other physicochemical properties and had significantly higher concentrations compared with the values obtained from boreholes located far distance away from Cemeteries. Turajo et al [23] findings were in support of earlier study by Gray et al. [24] in England that reported a reduction in contamination levels as the distance from cemeteries increases. In Benin City, Idehen [25] conducted univariate and multivariate analyses on the dataset obtained from characterization of groundwater samples from cemetery vicinity and identified Cl, $NO_3$, $SO_4$, $BOD_5$, Na, K, Mg, Pb, Mn, Cu, Ni, and Zn as parameters that could bear imprints of burial leakage from cemeteries. Trick et al. [26, 27] found significant concentrations of Cl, $SO_4$, $NO_3$, $HCO_3$, $CO_3$, Mg, K, Na and Ca and high temporal variation of $SO_4$, Na, Cl, and TOC content in water sampled from shallow groundwaters at a cemetery in Wolverhampton, England. Occasionally, the water samples were observed to contain high concentrations of organic carbon and ions including, $NH_4$, Cu, Mn, Zn, Fe, and As. Trang and Luan [6] in the study to determine impacts of longstanding cemetery on environment and solutions for urban cemetery planning in Ho Chi Minh City, Vietnam, observed that traditional interment has resulted in some widespread pollution of underground water due to leachates exuding from decay of corpses. The authors therefore recommended periodic assessment of the groundwater quality.

In Nigeria, different religious and cultural traditions support the practice of burying human remains. More often than not, cemeteries are sited close to settlements, in addition to cultural and religious reasons, as well as unavailability of land in populous areas. In many developing nations including Nigeria, cemeteries were typically built without conducting adequate geological and hydrological analyses, resulting in adverse effects on the environment and public health [28]. Also, most of the cemeteries in Nigeria are not provided with wellhead protection program contaminant-source inventories; hence the groundwater quality is often not monitored for possible contaminations from burial leak. The potential health risk that may result from the impact of cemeteries on groundwater quality is not often given enough attention in Nigeria [18]. Consequently, cemeteries are frequently situated close to settlements and within the influence of water sources since they have never been considered to pose a serious risk of environmental contamination [18].

Benin Metropolis in southern Nigeria obtains over 90% of its domestic and potable water from groundwater sources specifically from boreholes, which also serve for industrial and agricultural purposes [28]. Inadequate treatment of drinking water obtained from rivers has exposed the inhabitants of the semi urban and rural areas close to the metropolis to the health hazards associated with groundwater contaminants [29]. Like most urban regions in developing nations, the Benin metropolis sites large cemeteries adjacent to residential areas without taking into account the possible risk to the local population or environment [25, 30, 31]. This poses health risk as continued burying of the dead in limited space could result in high concentration of decomposition products which over time can endanger the groundwater thus making the groundwater unsuitable for any use.

Many indices have been established specifically for the purpose of assessing the health risk connected to diverse human activities [32]. The consumption of water contaminated by toxic substances, including heavy metals, has been linked to a wide range of public health problems, including vascular disease, restrictive lung disease, hypertension, cancer, gastrointestinal ulcers, neurological disorders, and reproductive failure [33–35]. To determine the total exposure to heavy metals among residents of a certain location, a health risk assessment of heavy metals can be performed [33]. In this regard, risk assessment of pollutants is either categorized as carcinogenic or non-carcinogenic to humans [36]. Typically, ingestion, inhalation, and dermal routes are the three main ways that we are exposed to chemicals in the various matrixes of our environment [32].

For effective evaluation of groundwater quality at sites close to the cemeteries, it would be essential to examine any possible human health impacts of the heavy metal concentrations in the groundwater. This is due to the fact that heavy metals are among the major contaminants at cemetery sites as highlighted in quite number of studies [9, 18, 25–28] and the water source is destined for drinking and other domestic purposes. Studies have shown that the traditional practice of estimating health impacts through direct comparison of analyzed levels with guideline limits is not effective in the assessment of comprehensive hazard levels and estimation of contaminants of most significance [37]. An essential method for analysing potential health impacts of pollutants in diverse dietary and environmental matrices is health risk assessment [38]. Health risk assessment of consumable and inhalable resources is a vital method to determine the potential impact of trace pollutants in water, food, and air on human health based on some established indices [39, 40]. The aim of conducting human health risk assessment is to minimize risks associated with consumption of contaminated water and food by adopting a number of techniques [39–42] which further ensures effective resource monitoring, management, and sustainability [40]. In general, four steps including hazard identification, dose-response assessment, exposure assessment, and risk characterization have been frequently used to assess human health risks [34]. Globally, health risk assessment has been used extensively to measure and quantify the levels of human exposure to toxic substances [34, 39–45].

Other aspects in the determination of water quality assessment involve adopting evaluation indices to evaluate the general quality of the water. It is challenging to evaluate the quality of water since various factors that influence the water's overall quality are taken into account [46]. However, the development of water quality evaluation indices reduces the data volume of multiple variables with diverse weights in overall quality of water to a great extent and simplifies the expression of water quality status [47]. Water quality evaluation indices are very important tools for examining and communicating raw or processed environmental information to stakeholders including decision-makers [48, 49]. In recent years, a multiplicity of water quality evaluation indices has been used to evaluate the general quality of water [25, 50–54].

This study was conducted to provide information on the impact of longstanding cemeteries on the quality of groundwater and associated health risk in Benin metropolis, Nigeria. To achieve this goal, we did the following (1) characterised the levels of the pH, electrical conductivity, alkalinity, chloride, phosphate, nitrate, sulphate, ammonium, calcium, sodium, potassium, biochemical oxygen demand, chemical oxygen demand, manganese, cadmium, nickel, lead, zinc, and iron which had been recommended by Tredoux et al. [55] as potential indicators of groundwater contamination of burial leak from cemetery in the water samples obtained across the study sites and compared their values against water quality regulatory standards, (2) adopt an index approach to determine the gross quality of the water aquifer reserves and further assess the health risks associated with oral and dermal consumption of water (3) explore the estimated physicochemical parameters to unveil the nature of the correlations among them via univariate and multivariate statistical methods. The specific objectives of this study were to (i) determine the levels, spatial variations and correlations of pollution potential indicators of burial leak and further compare the values against national and international regulatory standards such as the Nigerian Standard for Drinking Water Quality [56]; World Health Organization guidelines for drinking water quality [57] and Environmental Protection Agency Drinking Water Standards and Health Advisories [58]; (ii) determine the groundwater pollution status across the study sites using the parameter levels (iii) estimate health risks of non-carcinogenic metal such as manganese, copper, nickel, lead, zinc, and iron and carcinogenic metals (nickel and lead) via the daily drinking of the groundwater and dermal pathways for children and adults. We hypothesised that there are no correlations of pollution indicators of burial leak, and the variables are compatible with the national and international water quality

regulatory standards. Furthermore, that the buried leak poses no risk to human health or groundwater contamination. It is worthy to note that no previous studies have investigated the health risks associated with presence of carcinogenic and non-carcinogenic metals in groundwater resources upon which the populace within the locality of this study depend on to meet their daily demand for water. Our study is significant as it determines the extent to which the groundwater is impacted as previous study on litho-stratigraphic and hydrogeological [59] revealed that the materials overlaying the aquifer within Benin City are dominated by sands with traces of clay and lignite inter-bed and this condition has relatively low barrier to prevent infiltration of contaminants into aquifer. The results of our study would provide an important scientific understanding of the impact of longstanding cemeteries on groundwater quality and invariably argument valuable information for various stakeholders including policymakers and town/urban planning managers to implement appropriate remediation methods that would ensure the safety of groundwater resources. The uniqueness of our study is that it contributed evidence for realisation of Sustainable Development Goal 6 "Ensure availability and sustainable management of water and sanitation for all" in a fast-developing urbanised landscape by means of comprehensive water quality and health risk evaluation techniques.

## Materials and methods

### Study area

Benin City, located within the geographical coordinates; Northing 06˚ 20'21.0660" and Easting 05˚ 37' 2.8092" is the study area. Benin City covers an area of 748.35 km$^2$ in the southern part of Nigeria located within the tropical region which is characterized by two climatic regimes namely the wet season (April–October) and the dry season (November–March). The average annual rainfall is about 2500 mm and a temperature of 28˚C, with a mean annual relative humidity of 85% [60]. Two of the three cemeteries in the city were selected on the regularity of interment. These cemeteries included the Second and Third Cemeteries; the First Cemetery is not in use due to inaccessibility of the road network and poor drainage system. Consequently, the environment of Second Cemetery was excluded among the focus of this study. The cemeteries have existed for more than 50 years in Benin metropolis. The burial load of these cemeteries could not be ascertained due to inadequate record-keeping regarding the number of people buried, however, Idehen [25] observed that about 18–42 bodies of varying sizes are buried per week. This is further complicated where a single grave is used for multiple burials.

The geology of the area is characterized by the Benin Formation and built on a nearly undulating surface. Several studies have affirmed that this formation is underlain by sedimentary formation of the South Sedimentary Basin which follows the Oligocene-Pleistocene era in the continent of Africa and recent Oligocene-Pleistocene at the sub-oceanic [61–63]. The formation consists of top reddish earth composed of ferruginized or literalized clay sand, sand capping highly porous fresh water bearing loose pebbly sands, and sandstone with local thin clays and shale interbeds which are of braided stream origin [62]. Colours of the sands, sandstones and clays on weathered surfaces vary from reddish brown to pinkish yellow and to white in the deeper fresh surfaces. The brown reddish-yellowish colour is because of the limonitic coatings. Benin Region is drained by three river networks which include Ikpoba River, Ogba River and Owigie-Ogbovben River [62] while the hydrogeology study by Omorogieva and Imasuen [59] showed the aquifer underlying Benin region to be shallow flowing in direction of North-West to South-East from Northwest to Southeast and the overlaying materials dominated by sands with traces of clay and lignite inter-bed. Topography of this area is undulating in nature and has been described as a tilled plain in the southwestern direction as it is surrounded by the Benin historical moats [62, 64].

## Sample collection and analytical procedure

Permission to access and collect groundwater samples from boreholes in the privately-owned houses was granted verbally by all landlords in the study area". The landlords would however want to remain anonymous. Groundwater samples for evaluation of the physical and chemical parameters of the water were collected in 1liter polyethylene bottles on monthly interval for a period of five months (January to May, 2018) from outlet pipes of the borehole connected to the submersible pumping machine at each of the sample collection points. Samples were collected from six different boreholes located within 150 m from the cemetery sites. Five samples were collected from each of the 6 boreholes which aggregated to a total of 30 water samples from the two study sites. At each of the boreholes, the outlet pipe was swabbed with cotton wool soaked in 70% ethanol and water samples were collected after flushing for 5 minutes. All bottles were previously washed with dilute nitric acid (to remove heavy metals that might have been adsorbed to the wall of the container) and then rinsed with distilled water. At each of the collection point, the bottle was first rinsed with sample water before filling. The samples were labelled, then kept in a cooler box with ice and transported to Laboratory for Ecotoxicology and Environmental Forensics, University of Benin, Benin City, Nigeria for analysis within 48 hours [65, 66].

Hydrogen ion concentration (pH) and electrical conductivity (EC) were determined in-situ with a multiparametric probe (6920 V2-1 Multiparameter Water Quality Sonde, Xylem Analytics, USA). Otherwise stated, all the sample analyses were performed in compliance with the standard methods described by the American Public Health Association [67]. The alkalinity levels were determined by titrimetric methods using phenolphthalein and methyl orange as indicator. Phosphate, nitrate, ammonium nitrogen and sulphate were determined by a colorimetric method using a UV-visible spectrophotometer (DR 5000TM UV spectrophotometer) while MOHR's method in which silver nitrate is used as titrant and potassium chromate as the end point indicator was used to estimate concentrations of chloride. The levels of sodium and potassium were determined using flame photometer (Technicon Auto Analyzer flame photometer IV) while calcium was determined by direct complexation titration with ethylenediaminetetraacetic acid (EDTA) using Eriochrome Black T (EBT) indicator. With the aid of multiparametric probe, dissolved oxygen levels were estimated at the time of sample collection and on the fifth day after collection to determine the levels of biochemical oxygen demand ($BOD_5$). The levels of chemical oxygen demand (COD) were determined by titrimetric method using dichromate and ferrous ammonium sulphate.

Prior to the determination of the heavy metal concentrations, the water samples were digested using the acid digestion (Method 3005a) technique. The digest was diluted to 50 ml mark with distilled de-ionized water, and analyzed for iron (Fe), manganese (Mn), zinc (Zn), copper (Cu), cadmium (Cd), lead (Pb), Chromium (Cr) and nickel (Ni). The heavy metals in the solution were then read using an Atomic Absorption Spectrophotometer–AAS (Perkin–Elmer A3100). Prior to the analysis, calibration was done with a standard of known concentrations. The concentrations of Fe, Mn, Zn, Cu, Cd, Pb, and Ni were ascertained from the data generated by the AAS and expressed in mg $L^{-1}$.

Quality assurance method for heavy metal determination and quantification: Precision and accuracy of the AAS method was validated by repeating each procedure thrice. Certified reference documentations by the Federal Environmental Protection Agency [68] were used as a reference guide. The recovery rates which ranged from 92–96%, with relative standard deviations <5% indicate a high data integrity. The reference solutions for the calibration of curves were prepared from a stock solution containing 1000 mg $L^{-1}$ of each of the heavy metals analyzed. Similarly, the blanks and reference solutions were analysed using the same method that was

adopted for the samples and the concentrations were expressed in mg L$^{-1}$. The limits of detection (LoD—minimum detectable amount of the heavy metal), and limits of quantitation (LoQ–lowest measurable concentration of the heavy metal) were calculated based on the standard deviation of 12 readings obtained for the analytical blanks and the slopes of the analytical curves. LOD = 3σ/slope, and LOQ = 10σ/slope, where σ is the signal of the blank. The LoD values (mg L$^{-1}$) for the metals were, 0.00001 (Fe), 0.00001 (Mn), 0.00001 (Zn), 0.00001 (Cu), 0.0001 (Pb), and 0.00001 (Ni). The LoQ values (mg L$^{-1}$) were 0.0001 (Fe), 0.0001 (Mn), 0.0001 (Zn), 0.0001 (Cu), 0.001 (Pb), and 0.0001 (Ni).

## Statistical analysis

The data obtained were subjected to descriptive analysis and inter-station comparisons were conducted to test for significant differences in the various parameters characterized using parametric one-way analysis of variance (ANOVA). Using the months as replicates, ANOVAs were performed for each of the parameters. The data were log-transformed prior to the analysis and Shapiro-Wilk test was used to check for normality and homogeneity of variance and when normality was rejected ($p < 0.05$), Kruskal-Wallis test was used instead. Dunn's post hoc test was used to locate the source of significant difference across the stations. The relationships among the parameters were determined by Pearson correlation plot and Bonferroni adjusted p values were presented. These analyses were performed using PAST (version 4.03) software for windows. Principal component analysis (PCA) was performed using SPSS (version 26.0) software for windows to understand the hypothetical variations in the parameters. Varimax rotation method was used in the exploratory factor analysis due to its robustness to reduce the number of parameters with high eigenvectors for the various components [50]. The principal components (PCs) provide identification of the significant parameters from the whole dataset, thus offering data reduction with minimal loss of background information [69]. Similarly, the influence of less significant parameters generated from the dataset are reduced by the PCs as new groups of parameters are generated from PCA. The suitability of the data for the PCA were validated using Kaiser-Meyer-Olkin (KMO) and Bartlett's tests [70]. While KMO evaluates adequacy of the data, Bartlett's test measures the significance of associations between the parameters at a significant level. Eigenvalues were obtained for the 30 sets of samples through communality extractions and those ≥ 1 was adopted as the significant benchmark for the PC selection.

## Water quality evaluation indices

Water quality index (WQI), Heavy metal evaluation index (HEI), and Nemerow pollution index (NI) were investigated for the purpose of evaluating potability of the groundwater from the study areas. Generally, WQI is adopted for particular use of water and in this study, we considered the index for human consumption. WQI was computed using the weighted arithmetic index method as described by Cude [71] (Eq (1)) while using Nigerian Standard for Drinking Water Quality -NSDWQ [56]. The parameters adopted for the computation included pH, electrical conductivity, Cl$^-$, NO$_3$$^-$, SO$_4$$^{2-}$, PO$_4$$^{3-}$, Ca$^{2+}$, Na$^+$, BOD$_5$.

$$WQI = \frac{\sum_{i=1}^{n} WiQi}{\sum_{i=1}^{n} Wi} \tag{1}$$

Where, Wi = Relative weight of $n$th parameter, Qi = Quality rating of $n$th parameter.

$$Wi = \frac{1}{Si} \tag{2}$$

where Si = Standard permissible value for nth parameter, 1 = Proportionality constant.

$$Qi = \left\{ \left[ \frac{Vactual - Videal}{Vstandard - Videal} \right] *100 \right\} \qquad (3)$$

Where V actual = Actual value of the water quality parameter obtained from laboratory analysis.

V ideal = Ideal value is the neutral condition of the water quality parameter. V ideal for pH = 7 and for other parameters it is equal to zero.

V standard = values from NSDWQ [56] for respective parameters.

The quality rating and relative weight were calculated by standard procedure as described by Cude [71]. Water quality status can be described as excellent ($< 50$), good (50–100), poor (101–200), very poor (201–300) and unsuitable for domestic uses ($> 300$) [50]. HEI evaluates the overall quality of the water with respect to heavy metals [52] and calculated using the following equation.

$$HEI = \sum_{i=1}^{n} \frac{Hc}{Hmac} \qquad (4)$$

Where, *Hc* is analytical value of the heavy metal.

*Hmac* is maximum permissible concentration of the heavy metal according to NSDWQ [56] of the ith parameter and n = the number of parameters which was used in the computation.

The HEI values are categorized into three including low contamination HEI $\leq$10, medium contamination HEI (10–20) and high contamination HEI$>$20 [34, 53]. We applied NI to determine how different heavy metals characterised polluted groundwater at the study area [53] and determined by the equation.

$$NI = \left( \frac{\left[ \left( \frac{1}{n} \right) \sum (Ci/Si) \right]^2 + \left[ \max(Ci/Si) \right]^2}{2} \right)^{1/2} \qquad (5)$$

where, n = number of heavy metals used for the calculation, Ci = analytical value of the heavy metal i, Si = standard value according to NSDWQ [56] for the various heavy metals characterised. In line with NI, the degrees of heavy metal pollution of groundwater are divided into 6 which included no pollution: $\leq$0.5, clean: 0.5–0.7, warm: 0.7–1.0, polluted: 1.0–2.0, medium pollution: 2.0–3.0, and severe pollution$>$ 3.0 [53].

## Human health risk assessment

**Chronic risk assessment.** Risk assessment entails the procedures of estimating the probability of occurrence of any given likely magnitude of adverse health conditions over specified time period; it is a function of the hazard and exposure [25]. The health risk assessment of the heavy metal is usually quantified at various risk levels and expressed in terms of carcinogenic or non-carcinogenic health risk [34]. The two main toxicity risk factors evaluated are the reference dose (RfD) and slope factor (SF) which are, respectively used for non-carcinogen risk characterization and carcinogen risk characterization [72]. Human body is exposed to the contaminants via three main pathways including direct ingestion, dermal absorption and inhalation through mouth and nose [25]. For our study, oral and dermal exposure pathways were considered for the risk assessment. The chronic daily intake (CDI) of heavy metal through oral and dermal pathways were calculated by adopting methods of USEPA [42], Wu et al. [43] and

Karim et al. [44].

$$CDI_{oral} = \frac{(CW*IR*EF*ED)}{(BW*AT)}$$ (6)

$$CDI_{dermal} = \frac{(CW*SA*Kp*ET*EF*ED*CF)}{(BW*AT)}$$ (7)

where, $CDI_{oral}$ and $CDI_{dermal}$ indicate the exposure dose (mg/kg/day) through oral ingestion and dermal pathway, respectively and were calculated using the parameters value in Table 1.

Health risk assessment involves the characterization of potential adverse effects in reference to human health when exposed to contaminants [74]. Non-carcinogenic risks for exposure to heavy metals were evaluated by comparison of the estimated contaminant intakes from each exposure route with the reference dose (RfD) to express the hazard quotient (HQ) (Eq 8) [42]. When the value of HQ exceeds 1, an unacceptable risk of adverse non-carcinogenic effects on the health would be likely, while value is less than 1, it is an acceptable level [75, 76].

$$HQ = \frac{CDI}{RfD}$$ 8

where, HQ = hazard quotient (which is unit-less) and RfD is the reference dose (mg/Kg/day), recommended RfD for the metals are presented in Table 2.

To evaluate the overall potential for non-carcinogenic effects posed by more than one chemical, the HQs calculated for each chemical (heavy metals) are summated (Eq 9) and expressed as hazard index (HI) [42].

$$HI = \sum_{i=1}^{n} HQ = HQ_{Mn} + HQ_{Cd} + HQ_{Cu} + HQ_{Ni} + HQ_{Pb} + HQ_{Zn} + HQ_{Fe}$$ 9

If $HI > 1$ implies an unacceptable risk of non-carcinogenic health effects, while $HI < 1$ infer an acceptable level of risk (Table 3) [77].

*Carcinogenic risk assessment.* Carcinogenic risk (CR) was estimated as the incremental probability of an individual developing any form of cancer over lifetime due to twenty-four

**Table 1. Parameters for exposure assessment via oral ingestion and dermal absorption pathway.**

| Parameters | Unit | Oral values | Dermal values | References |
|---|---|---|---|---|
| CW (Conc. of heavy metal in water) | mg L$^{-1}$ | | | Study data |
| IR (Ingestion rate) | L day$^{-1}$ | 2.2 (Adult) | | [34, 43, 44] |
| | | 1 (Child) | | |
| EF (Exposure frequency) | days year$^{-1}$ | 365 | 350 | [34, 43, 44, 73] |
| ED (Exposure duration) | year | 70 (Adult) | 30 (Adult) | [34, 43, 44, 73] |
| | | 10 (Child) | 6 (Child) | |
| ET (Exposure time) | h event$^{-1}$ | | 0.58 (Adult) | |
| | | | 1.0 (Child) | |
| BW (Body weight) | kg | 70 (Adult) | 70 (Adult) | [34, 43, 44, 73] |
| | | 25 (Child) | 25 (Child) | |
| AT (Average time) | days | 25,550 (Adult) | 25,550 (Adult) | [27, 56–58] |
| | | 3,650 (Child) | 3,650 (Child) | |
| SA (Skin-surface area) | cm$^2$ | | 18,000 (Adult) | [27, 56–58] |
| | | | 6600 (Child) | |
| Kp (Permeability coefficient) | cm hr$^{-1}$ | | 0.001(Mn) (Cu) (Cd) (Fe), 0.0002 (Ni), 0.004 (Pb), 0.0006 (Zn) | [27, 56–58] |
| CF (Conversion factor) | L (cm$^3$)$^{-1}$ | | 0.001 | |

**Table 2. RfD and Slope factor of some metals.**

|    | RfD oral (mg/kg/day) | RfD dermal (mg/kg/day) | Oral SF (mg/Kg/day) |
|----|----------------------|------------------------|---------------------|
| Mn | $1.40 \times 10^{-1}$ | $8.00 \times 10^{-4}$ |  |
| Cd | $5.00 \times 10^{-4}$ | $5.00 \times 10^{-6}$ | 0.38 |
| Cu | $4.00 \times 10^{-2}$ | $1.20 \times 10^{-2}$ |  |
| Ni | $2.00 \times 10^{-2}$ | $5.40 \times 10^{-3}$ | 0.91 |
| Pb | $3.50 \times 10^{-3}$ | $5.25 \times 10^{-4}$ | 0.0085 |
| Zn | $3.00 \times 10^{-1}$ | $6.00 \times 10^{-2}$ |  |
| Fe | $7.00 \times 10^{-1}$ | $4.50 \times 10^{-2}$ |  |

Source: Bodrud-Doza et al. [34]; EPA [38]

hours per day exposure to a given daily amount of a carcinogenic substance for seventy years [78]. Carcinogenic risk index was calculated for Ni and Pb using Eq 10 [55]. Cd, Ni and Pb are the carcinogenic elements characterised in the samples, hence the CR was calculated only for these elements in this study.

$$CR = CDI_{oral} \times SF \qquad\qquad 10$$

where CR and SF denoted carcinogenic risks and Slope factor, respectively. The slope factors used for our study are presented in Table 2.

## Results and discussion

### Variations of the physico-chemical parameters

The summary of physicochemical parameters of groundwater samples from the study is presented Table 4 while the raw data for all the physico-chemical parameter are presented in S1 Table. The mean values of the pH of the groundwater samples varied from 3.83 to 4.80, indicating a highly acidic water body. Also, these mean values differed significantly (p <0.01) and the post hoc test nested the samples into two subgroups with reference to each of the cemeteries. None of the samples obtained falls within the permissible limits stipulated by national and international standards for drinking purposes. Idehen [25] and Alagbe et al. [22] reported slightly acidic pH conditions for groundwater samples collected boreholes in close proximity to cemeteries within urbanised areas in Nigeria. Contrary to our finding, Turajo et al [23] reported that by high pH values in a similar study. Low pH or acidic water had been noted for calcareous rocks of Benin, Mamu, Nsukka, and Ogwuasi-Asaba formations [79]. Electrical

**Table 3. Scales for chronic and carcinogenic risk assessment.**

| Risk level | HQ/HI | Chronic risk | Calculated cases of cancer occurrence | Cancer risk |
|------------|-------|--------------|----------------------------------------|-------------|
| 1 | <0.1 | Negligible | <1 per 1000,000 inhabitants ($10^{-6}$) | Very low |
| 2 | $\geq 0.1 < 1$ | Low | >1 per 1000,000 inhabitants ($10^{-6}$) | Low |
|   |  |  | <1 per 100,000 inhabitants ($10^{-5}$) |  |
| 3 | $\geq 1 < 4$ | Medium | >1 per 100,000 inhabitants ($10^{-5}$) | Medium |
|   |  |  | <1 per 10,000 inhabitants ($10^{-4}$) |  |
| 4 | $\geq 4$ | High | >1 per 10,000 inhabitants ($10^{-4}$) | High |
|   |  |  | <1 per 1000 inhabitants ($10^{-3}$) |  |
|   | - | - | >1 per 1000 inhabitants ($10^{-3}$) | Very high |

Source: Bodrud-Doza et al. [34]

**Table 4. Concentrations of physico-chemical parameters in the groundwater samples.**

| Parameters | Unit | B1 | B2 | B3 | B4 | B5 | B6 | p-value | NSDWQ [56] | WHO [57] | EPA [58] |
|---|---|---|---|---|---|---|---|---|---|---|---|
| | | $\bar{\times}$±SD; n = 5 | $\bar{\times}$±SD; n = 5 | $\bar{\times}$±SD; n = 5 | $\bar{\times}$±SD; n = 5 | $\bar{\times}$±SD; n = 5 | $\bar{\times}$±SD; n = 5 | | | | |
| pH | | 3.83 [b] ±0.34 | 4.05 [b] ±0.24 | 4.02 [b] ±0.22 | 4.80 [a] ±0.02 | 4.51 [a] ±0.21 | 4.52 [a] ±0.22 | 0.000** | 6.5–8.5 | 6.5–8.5 | 6.5–8.5 |
| EC | µS/cm | 237.99 [a] ±10.97 | 244.61 [a] ±8.70 | 246.00 [a] ±8.93 | 48.99 [b] ±2.23 | 40.05 [c] ±0.05 | 39.98 [c] ±0.01 | 0.000** | 1000 | - | - |
| Alkalinity | mg L⁻¹ | 0.20 [c] ±0.01 | 2.42 [b] ±3.28 | 3.99 [a b] ±2.41 | 4.19 [a] ±0.45 | 5.06 [a] ±1.00 | 4.98 [a] ±1.00 | 0.000** | - | - | - |
| Chloride | mg L⁻¹ | 46.58 [a] ±3.84 | 49.42 [a] ±7.07 | 52.24 [a] ±6.31 | 16.12 [b] ±0.02 | 14.14 [b] ±0.01 | 20.10 [a] ±0.01 | 0.000** | 250 | - | 250 |
| Phosphate | mg L⁻¹ | 0.27 [a] ±0.03 | 0.22 [a b] ±0.06 | 0.24 [a b] ±0.07 | 0.32 [a] ±0.38 | 0.23 [a b] ±0.07 | 0.18 [b] ±0.07 | 0.049* | - | - | - |
| Nitrate | mg L⁻¹ | 3.79 [a] ±0.77 | 4.14 [a] ±0.39 | 4.05 [a] ±0.30 | 0.79 [b] ±0.02 | 2.75 [a] ±0.71 | 2.63 [a] ±0.34 | 0.000** | 50 | 50 | 10 |
| Sulphate | mg L⁻¹ | 3.59 [a] ±0.54 | 2.81 [a] ±0.84 | 3.40 [a] ±1.14 | 1.97 [b] ±0.09 | 3.02 [a] ±0.01 | 2.98 [a] ±0.01 | 0.003** | 100 | - | 250 |
| Ammonia-N | mg L⁻¹ | 0.95 [a] ±0.35 | 0.75 [a] ±0.40 | 0.87 [a] ±0.32 | 0.18 [b] ±0.01 | 0.69 [a] ±0.16 | 0.65 [a] ±0.15 | 0.001** | - | - | 30 |
| Calcium | mg L⁻¹ | 13.46 [a] ±0.89 | 16.04 [a] ±3.00 | 18.00 [a] ±2.08 | 4.85 [b] ±0.09 | 2.87 [c] ±1.21 | 3.35 [c] ±1.22 | 0.000** | 200 | - | - |
| Sodium | mg L⁻¹ | 0.68 [a] ±0.08 | 0.79 [a] ±0.11 | 0.64 [a] ±0.08 | 0.43 [b] ±0.03 | 0.50 [b] ±0.07 | 0.48 [b] ±0.08 | 0.000** | 200 | 50 | 30–60 |
| Potassium | mg L⁻¹ | 0.76 [b] ±0.06 | 0.80 [b] ±0.04 | 0.79 [b] ±0.03 | 0.64 [c] ±0.04 | 0.91 [a] ±0.07 | 0.89 [a] ±0.08 | 0.000** | - | - | - |
| BOD₅ | mg L⁻¹ | 1.27 [d] ±0.94 | 3.01 [b c] ±0.84 | 3.52 [b] ±0.69 | 5.66 [a] ±0.12 | 1.98 [c d] ±0.91 | 2.36 [b c] ±0.80 | 0.000** | - | - | - |
| COD | mg L⁻¹ | 25.99 [a] ±3.06 | 22.01 [b] ±3.40 | 21.80 [b] ±3.51 | 23.88 [a b] ±0.34 | 27.62 [a] ±2.51 | 27.58 [a] ±2.50 | 0.004** | - | - | - |
| Mn | mg L⁻¹ | 0.05±0.01 | 0.05±0.03 | 0.06±0.03 | 0.07±0.02 | 0.06±0.01 | 0.03±0.02 | 0.243 | 0.2 | 0.1 | 0.05 |
| Cd | mg L⁻¹ | 0.00±0.00 | 0.00±0.00 | 0.00±0.00 | 0.00±0.00 | 0.00±0.00 | 0.00±0.00 | - | 0.003 | 0.003 | 0.005 |
| Cu | mg L⁻¹ | 0.40±0.14 | 0.33±0.23 | 0.28±0.28 | 0.15±0.12 | 0.53±0.20 | 0.38±0.21 | 0.097 | 1 | 2 | 1 |
| Ni | mg L⁻¹ | 0.04±0.05 | 0.06±0.02 | 0.06±0.03 | 0.03±0.02 | 0.05±0.01 | 0.02±0.00 | 0.104 | 0.02 | 0.07 | - |
| Pb | mg L⁻¹ | 0.19±0.18 | 0.17±0.15 | 0.18±0.09 | 0.11±0.07 | 0.17±0.03 | 0.12±0.04 | 0.814 | 0.01 | 0.01 | 0 |
| Zn | mg L⁻¹ | 0.63 [a] ±0.65 | 0.68 [a] ±0.51 | 0.78 [a] ±0.22 | 0.36 [a b] ±0.22 | 0.76 [a] ±0.22 | 0.09 [b] ±0.05 | 0.017* | 3 | 4 | 5 |
| Fe | mg L⁻¹ | 0.39 [c] ±0.11 | 0.36 [c] ±0.22 | 0.80 [b] ±0.12 | 0.18 [c] ±0.11 | 3.52 [a] ±0.82 | 3.08 [a] ±0.83 | 0.000** | 0.3 | 0.3 | 0.03 |

Groundwater samples from Third Cemetery were coded B1, B2 and B3 while B4, B5 and B6 were codes for samples from Second Cemetery, this is applicable to other tables with the same codes.

$\bar{\times}$±SD = pooled mean generated from values across the months per station ± standard deviation; post hoc = values with different superscripts (a > b > c > d) are significantly different (p < 0.05) while values with same superscript are not significantly different (p > 0.05). Significant differences

* = p<0.05

** = p<0.001.

conductivity (EC) is seen as an important parameter when evaluating groundwater quality at cemetery vicinity [22, 55]. Higher values of EC are indicative for high ionic strength and increased concentration of dissolved solids of the groundwater [34, 80]. The mean values of EC which varied from 39.98 µS/cm at B6 to 246.00 µS/cm at B3 were significantly different (p

<0.01) across the sites. Although high EC values were obtained for Third Cemetery, all values of EC were within the permissible limits.

The recorded concentrations of each of the anions, including chloride, sulphate, phosphate, and ammonia nitrogen differed significantly (p<0.05) in the groundwater samples. The sources of the differences were predominantly driven by the concentrations recorded at B4 which were either lesser or greater than the values for the other sites. With reference to the anions, the groundwater resources of the areas are not impacted as the values obtained were within the permissible levels. The concentrations of phosphate and nitrate recorded for this study were high when compared to the range of 0.28 to 0.37 mg $L^{-1}$ obtained by Imoisi et al. [81] in a study assessing the quality of groundwater of urban settlements resident close to dumpsites while in another study conducted at northern part of Edo State, Olaniran and Williams [82] reported elevated values of chloride (92.87–95.24 mg $L^{-1}$) and sulphate (34.40–44.40 mg $L^{-1}$) when compared to values obtained for this study. In a study that addressed the potential impact of cemeteries in Brazil, Pacheco et al. [9] reported nitrate concentration of 2.1 mg $L^{-1}$; this value is within the range of our recorded values for our current study. In a similar study, Migliorini [83] related high concentrations of nitrogenous products in the groundwater to be strongly correlated to the decomposition of corpse. The concentrations of alkali metals (sodium and potassium) and alkali earth metal (calcium) in the groundwater samples varied significantly (p<0.05) across the study area. The trend Ca>K>Na of concentration obtained for this study is in contrast from Na>Ca>K which was reported for the distribution of ions in groundwater of sedimentary formation of the South Sedimentary Basin in Nigeria [84]. The trend observed validated Fineza et al. [10] quantification of element composition of human body of an adult that Ca dominates the metal constituent, followed by K and then Na. Fiedler and Graw [85] explained that Ca is released as a result of saponification reactions in corpses and can percolate through the earth materials to contaminate the aquifer.

Index measures of organic contamination including $BOD_5$ and COD are key to conducting assessments on the impacts of cemeteries on groundwater [31]. The mean concentrations of $BOD_5$ (ranging from 1.27–5.66 mg $L^{-1}$) and COD (ranging from 21.80–27.62 mg $L^{-1}$) varied significantly (p<0.01) across the sample collection sites. The greater values of $BOD_5$ at site B4 suggests a higher biodegradability of earthen material at the location [86]. In contrast to $BOD_5$, COD was low at site B4 (23.88 mg $L^{-1}$) and high at the other sites ($\approx$ 27.50 mg $L^{-1}$), indicating that the condition at B4 were less favourable conditions for efficient microbial decomposition of human remains [87–89]. The $BOD_5$ values obtained across the study area were similar to the values reported by Turajo et al [23].

The concentration of the heavy metals slightly varied across sites except for Zn and Fe that were not significantly different. The mean concentration of Zn ranged from 0.09 mg $L^{-1}$ at B6 to 0.78 mg $L^{-1}$ at B3 while Fe varied from 0.18 mg $L^{-1}$ at B4 to 3.52 mg $L^{-1}$ at B5. While the mean concentrations of Pb and Fe at all the sites exceeded all the standard limits adopted for this study, the concentrations of Ni only exceeded the stipulated level for Nigerian Standard for Drinking Water Quality [56]. Although Ni is categorised as essential elements in the trace amounts, the toxicology of higher concentration of this element is well documented [90]. Higher concentrations of Ni may cause oxidative stress, and cancer [91]. Excessive exposure of human to zinc could lead to abdominal pain, vomiting, electrolyte imbalance, and acute renal failures [89]. Pb is not known to play any biological function in the body, but it is categorised as one of most harmful materials for humans [35]. Except Cd that was not detected in all the groundwater samples (and was excluded from the subsequent analyses), detectable concentrations of the other heavy metals are of concern considering the harm they can cause to humans after relatively long exposure (chronic toxicity) [92]. Among other parameters, Tredoux et al. [55] suggested the inclusion of Mn, Cd, Cr, Cu, Ni, Pb and Zn at areas with high risk of

groundwater contamination from burials leakage. When ranked in descending order, the mean concentrations of heavy metals were Zn > Fe > Cu > Pb > Mn > Ni > Cr for sites B1 to B4 and Fe > Zn > Cu > Pb > Mn > Ni > Cr for sites B5 and B6. The results of this study can aid environmental awareness campaigns among residents of the study area, urban planners and decision makers to effectively plan step to ameliorate the impacts of burial leak on the groundwater and sustainably manage the environment.

## Correlations of the physicochemical parameters

Results from the Pearson correlation indicated that most of the water quality parameters exhibited significant correlations which imply a significant relationship between the water sample sourcing and characterised parameters. The correlation plot indicated that the anion parameters, including chloride, nitrate, sulphate, phosphate, and ammonia-N with correlation coefficients of -0.803, -0.800, -0.644, -0.761, and -0.794, respectively contributed significantly ($p < 0.05$) to the pH of the groundwater samples (Fig 1). The significant and inverse nature of these correlations are indications that these ions are continuous released and this reduces the pH of the system thus making the water acidic. The EC of the groundwater was practically driven by the anions (chloride and nitrate associated with correlation coefficients of 0.986 and 0.665, respectively) and cations (calcium and sodium associated with correlation coefficients of 0.953 and 0.805, respectively) as significant positive correlations were observed among pairs of these variable (Fig 1). The anions were positively correlated; this indicates that they are of common origin. Correlation plot shows that Ca and Na exhibited positively significant correlation (0.639) hence, there is high tendency of the two elements originating from the same source. Furthermore, Ca showed positive correlations with all the anions (chloride, phosphate, nitrate, sulphate, and ammonia-N with correlation coefficient of 0.944, 0.421, 0.586, 0.198, and 0.404, respectively) characterised for this study. The results of the correlations, high concentrations of chloride, phosphate, nitrate, sulphate, and ammonia-N, and Tredoux et al. [55] inclusion of these ions among the priority chemical constituents in assessing the impact of decomposition of human remains on groundwater, could be inferred that the aquifers of the

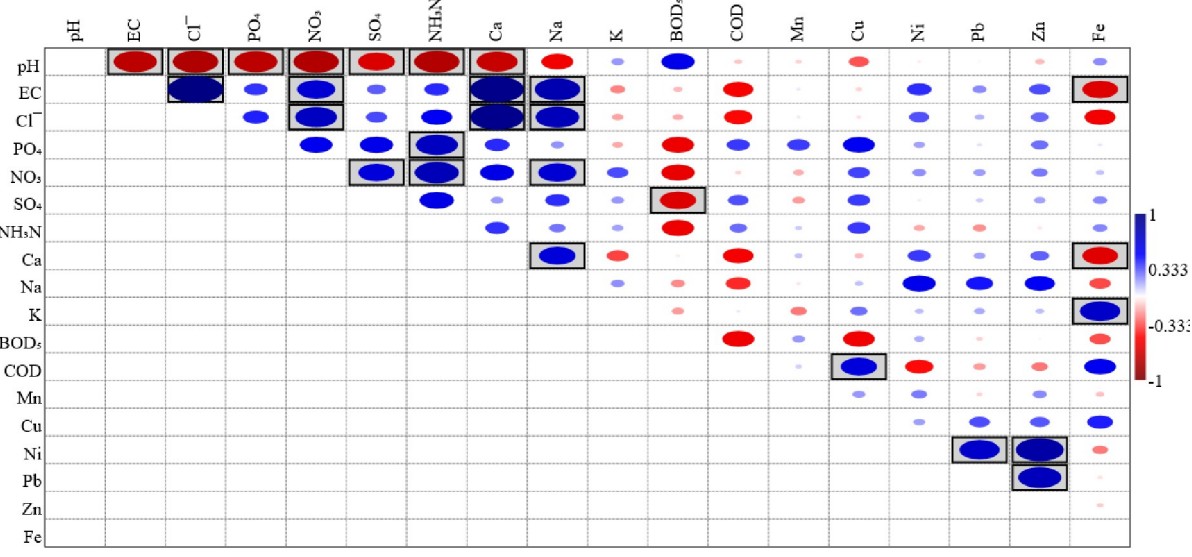

**Fig 1. Pearson correlation plot for the studied parameters of groundwater.** Highlighted boxes showed significant correlations at the 0.05 level (2-tailed).

study area are subjected to contamination from burials leakage. The inverse correlation between $BOD_5$ and COD contradicts other studies on groundwater quality [88–90]. Pairs of the heavy metals exhibited positive correlations except in conjugation with Fe whereas the correlations were significant for Ni versus Pb and Zn; Pb versus Zn. The correlations among the parameters generally implied the similar influence of the parameters by external factor(s). Meanwhile positive correlation infers common originality of parameters which may occur independent of one another. In the case of this study, common originality could be considered priority in reference to Tredoux et al. [55] records. In general, high levels of correlations observed among parameters characterised in the groundwater support the findings on litho-stratigraphic and hydrogeological study by Omorogieva and Imasuen [59] which revealed that the materials overlaying the aquifer within Benin City are dominated by sands with traces of clay and lignite inter-bed and this condition has enhances infiltration of contaminants into aquifer.

## Principal component analysis (PCA)

PCA was applied to describe the experiential interrelatedness of clusters of parameters in simple patterns; this was expressed in the patterns of variance and covariance between the parameters and similarities among observations [34, 93]. In this study, the KMO value of 0.505 indicates that PCA can be used; similarly, Bartlett's test with the p value < 0.001 implies that we could advance with the analysis. Results of the PCA showed that the first five principal components (PC) recorded eigenvalues $\geq 1$ and were used to explain the interrelationships among the parameters (Table 5). Fig 2 shows the three-dimensional plot of the PCA. The

**Table 5. Varimax rotated principal component analysis for groundwater samples.** Values in bold indicate most important variables per axis.

| Parameters | PC 1 | PC 2 | PC 3 | PC 4 | PC 5 |
|---|---|---|---|---|---|
| pH | **-0.770** | -0.592 | 0.061 | 0.158 | -0.042 |
| EC | **0.938** | 0.013 | 0.218 | -0.248 | -0.052 |
| Alkalinity | -0.440 | -0.315 | -0.254 | 0.491 | 0.532 |
| Chloride | **0.975** | 0.049 | 0.126 | -0.152 | -0.017 |
| Phosphate | 0.400 | **0.759** | 0.086 | -0.127 | 0.387 |
| Nitrate | **0.771** | 0.446 | 0.112 | 0.384 | -0.119 |
| Sulphate | 0.339 | **0.729** | 0.054 | 0.088 | -0.333 |
| Ammonium-N | **0.600** | **0.617** | -0.292 | 0.291 | 0.190 |
| Ca | **0.906** | -0.044 | 0.152 | -0.292 | 0.100 |
| Na | **0.724** | 0.067 | 0.510 | 0.052 | -0.260 |
| K | -0.065 | 0.063 | 0.183 | **0.923** | -0.219 |
| $BOD_5$ | -0.145 | **-0.837** | 0.038 | -0.118 | 0.271 |
| COD | -0.505 | **0.795** | -0.254 | 0.017 | 0.101 |
| Mn | 0.018 | 0.051 | 0.131 | -0.178 | **0.888** |
| Cu | -0.124 | **0.757** | 0.352 | 0.243 | 0.242 |
| Ni | 0.262 | -0.157 | **0.891** | 0.011 | 0.176 |
| Pb | 0.016 | 0.064 | **0.896** | 0.025 | -0.153 |
| Zn | 0.188 | 0.075 | **0.905** | 0.020 | 0.143 |
| Fe | -0.434 | 0.390 | -0.109 | **0.766** | 0.024 |
| Eigenvalue | 5.780 | 4.245 | 3.191 | 2.239 | 1.730 |
| % of Variance | 30.422 | 22.341 | 16.794 | 11.783 | 9.103 |
| Cumulative % | 30.422 | 52.763 | 69.557 | 81.340 | 90.442 |

Bold figures: Significant loading

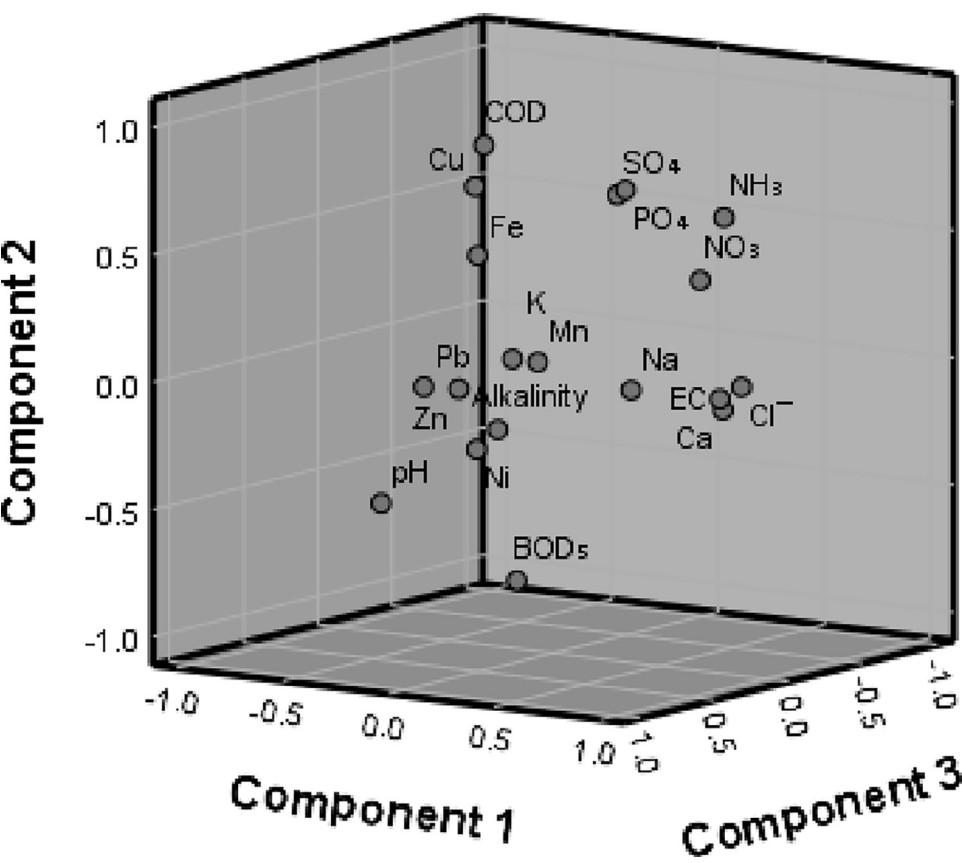

**Fig 2. Principal component analysis plot in rotated space.**

components PC1, PC2, PC3, PC4, and PC5 accounted for 30.422%, 22.341%, 22.341%, 11.783% and 9.103% of variance, respectively. With the exception of alkalinity, other parameters were found to score eigenvectors ($\geq$ 60%) for various PCs. Ammonium-N contributed significantly to PC 1 and 2, and together with pH, EC, chloride, nitrate, Ca, and Na, the significance was observed for PC 1 while with phosphate, sulphate, $BOD_5$, COD, and Cu for PC 2. The parameters highlighted for PC 1 and 2 have been denoted by different authors while addressing the impacts of burials leakage on groundwater quality [13, 25, 31, 55]. PC 3 was mainly derived by heavy metals (Ni, Pb, Zn). Rodrigues [13] reported Zn and Pb as the primary contaminants of groundwater samples from cemeteries. Heavy metals, including Fe, Cu, Pb, and Zn are among the constituents of wood preservatives and paints frequently used in the construction of coffins [9]. In addition, evidence of differences in the origin of heavy metal from burial decomposition to contamination of the groundwater can be viewed from a perspective that their high attributes were not pooled into a particular PC. For instance, Fe recorded high eigenvalue for PC 4 and this variation could be attributed to the major source of Fe which is order than materials used for coffin construction compared to the body of a 70 kg adult which contains approximately 4.2 g of Fe [7].

### Water quality assessment

The result of pollution evaluation indices is presented in Table 6. The ranges of WQI, HEI, HPI and NI were 170.91 to 234.54, 14.37 to 31.65, and 8.28 to 13.98, respectively. Water quality index provides a single value that indicates the water quality levels of a given source of water

**Table 6. Results of groundwater pollution evaluation indices.**

| Samples | WQI | WQI levels | HEI | HEI level | NI | NI levels |
|---|---|---|---|---|---|---|
| B1 | 194.50± 2.884 | Poor | 23.67± 20.434 | High contamination | 13.98± 13.207 | Severe pollution |
| B2 | 234.54± 3.397 | Very poor | 14.37± 8.044 | Medium contamination | 8.28± 4.976 | Severe pollution |
| B3 | 220.72± 20.738 | Very poor | 22.42± 16.641 | High contamination | 12.78± 10.650 | Severe pollution |
| B4 | 170.91± 8.851 | Poor | 31.65± 4.275 | High contamination | 12.61± 1.795 | Severe pollution |
| B5 | 233.09± 21.156 | Very poor | 24.28± 9.761 | High contamination | 13.05± 6.373 | Severe pollution |
| B6 | 178.42± 8.349 | Poor | 23.28± 5.455 | High contamination | 9.06± 3.062 | Severe pollution |

and enables easy interpretation and water quality assessments [94, 95]. While employing Ramakrishniah et al. [51] for classification of WQI values, the groundwater quality across the two locations was generally poor. Generally, the quality rating revealed that the poor quality of the groundwater across the areas was the resultant effects of pH and $BOD_5$. Knight and Dent [96] discovered high level of $BOD_s$ in groundwater obtained beneath the cemetery in Adelaide, Australia. The results of the heavy metal evaluation index (HEI) showed varying levels of contamination of the groundwater by heavy metals. Laying credence to Bodrud-Doza et al. [53] HEI scales of evaluation, all the groundwater samples showed high contamination level except B2 which indicated low contamination level. The results of Nemerow pollution index (NI) further showed that groundwater of the study area is severely polluted by heavy metals. The values of HEI and NI were practically driven by the levels of Pb in the groundwater samples. The values of variance for HEI were high, thus showing uneven release of heavy metal to groundwater resources at the areas. High value of variances is an indication for pollution which could be attributed to either or both natural and anthropogenic activities [97]. Generally, the different indices adopted in this study showed the importance of the various parameters highlighted by Tredoux et al. [55]. Our results showed incidents of organic pollution and heavy metal contamination of the groundwater [50, 95] which are potential impacts from burials leakage [23, 24]. This finding of our research study further portrays the low capacity of the overlaying earth materials (dominated by sands) to prevent infiltration of contaminants [59]. Trang and Luan [6] reported incident of heavy metal contamination in groundwater sampled from area of two longstanding cemeteries in Ho Chi Minh City. In another study, Rodrigues and Pacheco [98] emphasized a condition of Zn and Pb contamination of groundwater obtained from Luz de Tavira, Querenc and Seixas cemeteries in Portuguese.

## Human health risk assessment

Details from evaluation of chronic and carcinogenic risk are presented in Tables 7 to 9. Daily dosage consumption of the elements, hazard quotients and hazard index were explored to determine the contribution of the various heavy metals we characterised.

**Chronic risk.** Tables 7 and 8 show the summaries for human chronic health risk (HQ and HI) of groundwater through the exposure of oral and dermal routes, for adults and children. We found that HQ values for Mn, Cu, Ni, and Fe to be less than 1 for both children and adults. Our result suggests that these metals could pose low degree of health-related [76]. However, high HQ oral values (greater than 1) were found for Pb for children and adults and these values were the principal factor behind the overall HI values across the study area. In line with EPA [58], the samples suggest medium category of chronic risk for adults and 33.33% samples were found in high risk for children (Table 7). The high-risk area included samples from Second Cemetery (B5) and Third Cemetery (B1, B2, and B3).

Similar to our findings, Bodrud-Doza et al. [34] reported that the dermal exposure has comparatively lower risk than oral exposure for both children and adults. For the dermal route of

**Table 7. HQ and HI value for children and adult through oral exposure pathway.**

|  | Sample ID | HQ | | | | | | HI | Chronic risk according to USEPA [99] |
|---|---|---|---|---|---|---|---|---|---|
|  |  | Mn | Cu | Ni | Pb | Zn | Fe |  |  |
| Children | B1 | 0.0146 | 0.4019 | 0.0878 | 3.8706 | 0.0843 | 0.0209 | 4.4802 | High |
|  | B2 | 0.0150 | 0.3321 | 0.1258 | 3.4551 | 0.0901 | 0.0203 | 4.0383 | High |
|  | B3 | 0.0174 | 0.2825 | 0.1104 | 3.6017 | 0.1041 | 0.0456 | 4.1618 | High |
|  | B4 | 0.0191 | 0.1511 | 0.0690 | 2.2897 | 0.0474 | 0.0100 | 2.5864 | Medium |
|  | B5 | 0.0170 | 0.5345 | 0.0910 | 3.3125 | 0.1007 | 0.2010 | 4.2568 | High |
|  | B6 | 0.0096 | 0.3758 | 0.0308 | 2.3337 | 0.0122 | 0.1760 | 2.9380 | Medium |
|  | **Mean** | **0.0155** | **0.3463** | **0.0858** | **3.1439** | **0.0731** | **0.0790** | **3.7436** | **Medium** |
|  | B1 | 0.0115 | 0.3158 | 0.0690 | 3.0412 | 0.0663 | 0.0164 | 3.5201 | Medium |
| Adult | B2 | 0.0118 | 0.2609 | 0.0988 | 2.7147 | 0.0708 | 0.0160 | 3.1729 | Medium |
|  | B3 | 0.0137 | 0.2220 | 0.0867 | 2.8299 | 0.0818 | 0.0358 | 3.2700 | Medium |
|  | B4 | 0.0150 | 0.1187 | 0.0542 | 1.7991 | 0.0373 | 0.0079 | 2.0322 | Medium |
|  | B5 | 0.0134 | 0.4200 | 0.0715 | 2.6027 | 0.0791 | 0.1580 | 3.3447 | Medium |
|  | B6 | 0.0075 | 0.2953 | 0.0242 | 1.8336 | 0.0096 | 0.1383 | 2.3085 | Medium |
|  | **Mean** | **0.0121** | **0.2721** | **0.0674** | **2.4702** | **0.0575** | **0.0621** | **2.9414** | **Medium** |

exposure, the HQ values for all the samples for both children and adults were less than 1 and the HI grading of the samples was low and negligible for the children and adult, respectively (Table 8). In all the cases, the potential highest chronic risk was recorded for Pb while the potentially lowest and highest chronic risk across the sites for both children and adults were found in sampling sites B4 and B1, respectively for both oral and dermal routes.

## Carcinogenic risk

The risk of cancer in humans can be enhanced by some heavy metals including, Pb, Cr (VI), Cd, and Ni [100]. Long-term exposure to low amounts of these metals could result in different manifestations of cancer [33]. The result for carcinogenic risk assessment of the groundwater samples in relation to Ni and Pb for adults is presented in Table 9. With reference to Pb, all the

**Table 8. HQ and HI value for children and adult through dermal exposure pathway.**

|  | Sample ID | HQ | | | | | | HI | Chronic risk according to USEPA [99] |
|---|---|---|---|---|---|---|---|---|---|
|  |  | Mn | Cu | Ni | Pb | Zn | Fe |  |  |
| Children | B1 | 0.0097 | 0.0051 | 0.0002 | 0.2240 | 0.0010 | 0.0012 | 0.2412 | Low |
|  | B2 | 0.0100 | 0.0042 | 0.0004 | 0.1999 | 0.0010 | 0.0012 | 0.2167 | Low |
|  | B3 | 0.0116 | 0.0036 | 0.0003 | 0.2084 | 0.0012 | 0.0027 | 0.2278 | Low |
|  | B4 | 0.0127 | 0.0019 | 0.0002 | 0.1325 | 0.0005 | 0.0006 | 0.1485 | Low |
|  | B5 | 0.0113 | 0.0068 | 0.0003 | 0.1917 | 0.0011 | 0.0119 | 0.2231 | Low |
|  | B6 | 0.0064 | 0.0048 | 0.0001 | 0.1350 | 0.0001 | 0.0104 | 0.1568 | Low |
|  | **Mean** | **0.0103** | **0.0044** | **0.0002** | **0.1819** | **0.0008** | **0.0047** | **0.2024** | **Low** |
|  | B1 | 0.0039 | 0.0021 | 0.0001 | 0.0904 | 0.0004 | 0.0005 | 0.0973 | Negligible |
| Adult | B2 | 0.0040 | 0.0017 | 0.0001 | 0.0807 | 0.0004 | 0.0005 | 0.0874 | Negligible |
|  | B3 | 0.0047 | 0.0014 | 0.0001 | 0.0841 | 0.0005 | 0.0011 | 0.0919 | Negligible |
|  | B4 | 0.0051 | 0.0008 | 0.0001 | 0.0535 | 0.0002 | 0.0002 | 0.0599 | Negligible |
|  | B5 | 0.0046 | 0.0027 | 0.0001 | 0.0773 | 0.0005 | 0.0048 | 0.0900 | Negligible |
|  | B6 | 0.0026 | 0.0019 | 0.0000 | 0.0545 | 0.0001 | 0.0042 | 0.0633 | Negligible |
|  | **Mean** | **0.0041** | **0.0018** | **0.0001** | **0.0734** | **0.0003** | **0.0019** | **0.0816** | **Negligible** |

Table 9. Cancer risk of nickel (Ni), and lead (Pb) for adult via oral exposure pathway.

| Sample ID | Cancer risk values | | Carcinogenic risk according to USEPA [99] | |
|---|---|---|---|---|
| | Ni | Pb | Ni | Pb |
| B1 | 0.001256 | 0.000052 | Very high | Medium |
| B2 | 0.001799 | 0.000046 | Very high | Medium |
| B3 | 0.001579 | 0.000048 | Very high | Medium |
| B4 | 0.000987 | 0.000031 | High | Medium |
| B5 | 0.001301 | 0.000044 | Very high | Medium |
| B6 | 0.000440 | 0.000031 | High | Medium |
| Mean | 0.001227 | 0.000042 | Very high | Medium |

samples posed medium carcinogenic risk suggested by USEPA [99] while for Ni, 66.667% of the samples pose very high carcinogenic risk for the populace. The sampling sites which were most threatened due to Ni and Pb were found to be B2 and B1, respectively while the least threatened sampling sites were B6 for Ni and B4 for Pb.

Although some studies have evaluated the quality of water in Benin City [25, 99–104], health risk assessment of groundwater contamination by heavy metal has rarely been a focus of study in this area. The results obtained from the previous especially in respect to groundwater quality in Benin City and elsewhere in Nigeria were in consonance with our findings. For instance, Chinye-Ikejiunor et al. [103] reported abnormal concentrations of Pb, Cr, Zn and Cd in groundwater samples obtained from Onitsha, southeastern Nigeria while Enuneku et al. [104], results showed enrichment of heavy metals (Mn, Zn and Cu) in groundwater samples obtained from various borehole sites within the city. Idehen [25] reported marginal concentrations of nickel and lead in water samples obtained from Third Cemetery. Omoigberale et al. [105] discovered that iron and lead concentrations in groundwater sampled from Benin City to be above the WHO limits. At Lagos, a coastal city in Nigeria, Alagbe et al. [22] reported elevated levels of lead in groundwater samples obtained around Ayobo Cemetery. Our results went beyond usual comparison of values of the water quality parameters with regulatory standards to provide evidence that nickel concentrations in the groundwater pose high health risk to the populace. Mere comparison of Ni concentration with standard showed that the levels were compatible with WHO [57] regulatory standards for drinking water. Genchi et al. [106] gave details of side effects of Ni on human health to include allergy, cardiovascular, kidney diseases, lung fibrosis as well as lung and nasal cancer. Considering the important scientific evidence provided in this study, it is expedient for the stakeholders to implement appropriate remediation methods that would ensure the safety of groundwater resources or provide a safe alternative water to meet the domestic demand for the populace.

## Limitations of this study, recommendations and perspectives for future research

The results from our study showed that the burial leakages from longstanding cemeteries in Benin City are creating significant imprints on physicochemical characteristics of water in aquifer reserve therein. Thus, consumption as well as use of the groundwater from these sites for domestic, agricultural and recreational purpose without pre-treatment by physical or chemical or biological process could have severe impact on the health of populace.

One of the major limitations of this research study was that we collected groundwater samples from boreholes within the reach of 150m, thus our result cannot account condition beyond this distance. This condition was orchestrated by the limited funds available for this

study. Also, absent of background data on the physicochemical condition of the groundwater and records of bodies buried in the various cemeteries limited us from modelling the extent of the contamination of the system as well as projecting for the future conditions.

In order to complement the results from this study and further ensure the attainment of Goal 6 of SDGs, the following recommendations and future studies are advised.

- We recommend complementary study to understand the impact of burial leakage on the microbial composition in the groundwater and another to address the extent of the impact both on the physicochemical characteristic and microbial composition.

- The government should make haste to provide another site for interment of human remains and stop such activity at Second and Third Cemeteries. Selection of new site should be based on scientific evidence including prioritising the rate of city sprawling as well as lithological and hydrogeological characteristic (area with high clay content is preferable).

- The general public should be briefed of the contaminated groundwater and restriction should be placed on the uses of the water except otherwise treated. Means to ameliorate the impact the impact of the decision such as provision of alternative water source in adequate quantity and quality should be prioritised.

## Conclusion

Water is fundamental to every living thing, hence access to good quality can never be overemphasised. The results of our assessment of the groundwater quality and associated health risk for adjoining human settlements at cemeteries in Benin City, Nigeria, revealed that the groundwater from the study area is of poor quality, and highly contaminated by heavy metals. The modelled hazard quotient (HQ) and hazard index (HI) found that for oral exposure, approximately 33% of samples suggest the high category of chronic risk for children while medium category was indicated for adults. However, the carcinogenic risk of Ni and Pb via oral exposure route suggest very high risk for Ni and medium risk for Pb. In consideration that long term exposure to low concentrations of some heavy metals (including Pb, Cd, and Ni) could result in different manifestations of cancer, we recommend that another cemetery should be provided while government should provide the residents of these areas an alternative source of water for drinking and other domestic uses.

## Supporting information

**S1 Table. Physicochemical parameters data obtained from boreholes samples near cemeteries in Benin metropolis.**
(CSV)

## Acknowledgments

The authors thank their respective institutions for giving them the time to conduct this research and write the resultant manuscript.

## Author Contributions

**Conceptualization:** Ifeanyi Maxwell Ezenwa, Michael Omoigberale, Rachel Abulu, Ekene Biose, Benjamin Okpara, Osariyekemwen Uyi.

**Data curation:** Ifeanyi Maxwell Ezenwa, Michael Omoigberale.

**Formal analysis:** Ifeanyi Maxwell Ezenwa, Ekene Biose, Benjamin Okpara.

**Investigation:** Ifeanyi Maxwell Ezenwa, Rachel Abulu, Benjamin Okpara.

**Methodology:** Michael Omoigberale, Rachel Abulu, Ekene Biose, Benjamin Okpara.

**Project administration:** Ifeanyi Maxwell Ezenwa, Michael Omoigberale.

**Resources:** Michael Omoigberale.

**Supervision:** Ifeanyi Maxwell Ezenwa, Michael Omoigberale, Ekene Biose.

**Validation:** Ifeanyi Maxwell Ezenwa, Michael Omoigberale, Rachel Abulu.

**Visualization:** Ifeanyi Maxwell Ezenwa, Michael Omoigberale, Rachel Abulu, Benjamin Okpara.

**Writing – original draft:** Ifeanyi Maxwell Ezenwa, Michael Omoigberale, Rachel Abulu, Ekene Biose, Osariyekemwen Uyi.

**Writing – review & editing:** Ifeanyi Maxwell Ezenwa, Michael Omoigberale, Rachel Abulu, Ekene Biose, Benjamin Okpara, Osariyekemwen Uyi.

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
