## [Decision Letter · Decision Letter 0]

18 Jun 2023

PONE-D-23-11494Burial leakage: a human accustomed groundwater contaminant sources and health hazards study near cemeteries in Benin City, NigeriaPLOS ONE

Dear Dr. Uyi,

Thank you for submitting your manuscript to PLOS ONE. After careful consideration, we feel that it has merit but does not fully meet PLOS ONE’s publication criteria as it currently stands. Therefore, we invite you to submit a revised version of the manuscript that addresses the points raised during the review process.

We look forward to receiving your revised manuscript.

Kind regards,

Venkatramanan Senapathi, Ph.D.

Academic Editor

PLOS ONE

Groundwater pollution by trace metals and human health risk assessment in central west part of Bangladesh - https://doi.org/10.1016/j.gsd.2019.100219

Targeting low-arsenic aquifers in Matlab Upazila, Southeastern Bangladesh - http://dx.doi.org/10.1016/j.scitotenv.2006.06.028

In your revision ensure you cite all your sources (including your own works), and quote or rephrase any duplicated text outside the methods section. Further consideration is dependent on these concerns being addressed.

“NO - Include this sentence at the end of your statement: The funders had no role in study design, data collection and analysis, decision to publish, or preparation of the manuscript.”

d) If you did not receive any funding for this study, please state: “The authors received no specific funding for this work.

6. We note that Figure 1 in your submission contain [map/satellite] images which may be copyrighted. All PLOS content is published under the Creative Commons Attribution License (CC BY 4.0), which means that the manuscript, images, and Supporting Information files will be freely available online, and any third party is permitted to access, download, copy, distribute, and use these materials in any way, even commercially, with proper attribution. For these reasons, we cannot publish previously copyrighted maps or satellite images created using proprietary data, such as Google software (Google Maps, Street View, and Earth). For more information, see our copyright guidelines: http://journals.plos.org/plosone/s/licenses-and-copyright.

7. Please upload a copy of Supporting Information Table S1 which you refer to in your text on page 17.

8. In your Data Availability statement, you have not specified where the minimal data set underlying the results described in your manuscript can be found. PLOS defines a study's minimal data set as the underlying data used to reach the conclusions drawn in the manuscript and any additional data required to replicate the reported study findings in their entirety. All PLOS journals require that the minimal data set be made fully available. For more information about our data policy, please see http://journals.plos.org/plosone/s/data-availability.

Additional Editor Comments:

The authors have improved the manuscript, but not thoroughly enough.

There is a lot of information, but the clarity, coherence, and synthesis are still insufficient, as the reviewers pointed out.

Reviewers' comments:

Reviewer's Responses to Questions

**Comments to the Author**

1. Is the manuscript technically sound, and do the data support the conclusions?

Reviewer #1: Yes

Reviewer #2: Yes

2. Has the statistical analysis been performed appropriately and rigorously? 

Reviewer #1: Yes

Reviewer #2: Yes

3. Have the authors made all data underlying the findings in their manuscript fully available?

Reviewer #1: Yes

Reviewer #2: Yes

4. Is the manuscript presented in an intelligible fashion and written in standard English?

Reviewer #1: Yes

Reviewer #2: Yes

5. Review Comments to the Author

Reviewer #1: Comments:

1. “Ethics statement” title in “Materials and Methods” should be transferred to the end of conclusion

2. Add parentheses for “(Fig. 1: the map was plotted using QGIS Desktop 3.16.3 .”

3. Use a uniform format for “mgL-1” or “mgl-1” in the text.

4. In equation 10, use multiplication sign “x” correctly.

5. Use comma before “respectively”.

6. Table 4, row 3, columns 11 and 12, “-” is given in red color.

7. Add limitations of your study.

8. Compare your results with more from literature.

9. To improve the paper’s Introduction regarding water quality and metals, the authors can read and use the following papers (if applicable):

I. Distribution, exposure, and human health risk analysis of heavy metals in drinking groundwater of Ghayen County, Iran

II. Protocol for the estimation of drinking water quality index (DWQI) in water resources: Artificial neural network (ANFIS) and Arc-Gis

III. Characteristics, water quality index and human health risk from nitrate and fluoride in Kakhk city and its rural areas, Iran

Reviewer #2: Reviewer # comments:

Manuscript number: PONE-D-23-11494

Manuscript title: Burial leakage: a human accustomed groundwater contaminant sources and health hazards study near cemeteries in Benin City, Nigeria

Dear Authors, I have now read and assessed your manuscript with the abovementioned details. The topic of the manuscript is interesting. However, I have some concerns that should be carefully addressed. Please, see my queries/specific comments below.

1. Please sufficiently justify that the selected parameters would provide signatures of leachates that are peculiar to cemeteries. Check whether it is really essential to emphasize on the impact of cemetery.

2. Please emphasize more on the gaps in knowledge and novelty of the present study in the introduction section.

3. This statement is unclear: “Several studies affirmed that this formation which follows the Oligocene-Pleistocene era in the continent of Africa at the sub-oceanic is underlain by sedimentary formation of the South Sedimentary Basin.” Please refine to make geologic sense.

4. “Benin formation” should be written as “Benin Formation”. The norm is that the word “formation” should be capitalized.

5. Please provide sufficient details on the surface hydrology, topography, and hydrogeology. These are very important for a better understanding of the influence of cemetery on the water quality.

6. Geologic/hydrogeologic map is missing.

7. The results and discussion section should be robust. Compare your findings with literature data from other regions.

8. Please refer to these recent articles to improve the literature review, methodology and discussions: https://doi.org/10.1007/s13201-020-01180-9;
https://doi.org/10.1080/15275922.2021.2006363;
https://doi.org/10.1007/s10661-022-09789-w;
https://doi.org/10.1080/10106049.2022.2034990;
https://doi.org/10.1016/j.heliyon.2023.e15483;
https://doi.org/10.1016/j.ecolind.2023.110287;
https://doi.org/10.1007/s11356-023-26396-5

9. The quality of the figures is poor. Please replace them with better versions.

10. The novelty and significances of the current study should also be justified in the conclusions.

11. What are the limitations of the present study? Highlight on them.

12. Also, discuss the perspectives for future research.

Best wishes.

6. PLOS authors have the option to publish the peer review history of their article (what does this mean?). If published, this will include your full peer review and any attached files.

Reviewer #1: No

Reviewer #2: No

---

## [Author Response · Author response to Decision Letter 0]

9 Aug 2023

Reviewer #1: Comments:

1. “Ethics statement” title in “Materials and Methods” should be transferred to the end of conclusion

The Ethics statement have been transferred to the end of conclusion.

2. Add parentheses for “(Fig. 1: the map was plotted using QGIS Desktop 3.16.3 .”

On the recommendation of the journal, we have now deleted Figure 1

3. Use a uniform format for “mgL-1” or “mgl-1” in the text.

This has been unified throughout the manuscript.

4. In equation 10, use multiplication sign “x” correctly.

Done, thanks you.

5. Use comma before “respectively”.

This has been done throughout the manuscript.

6. Table 4, row 3, columns 11 and 12, “-” is given in red color.

The colour effect has been addressed. 

7. Add limitations of your study.

We have added limitations of the research study to the manuscript.

8. Compare your results with more from literature.

We have done this by comparing with more literature across the globe

9. To improve the paper’s Introduction regarding water quality and metals, the authors can read and use the following papers (if applicable):

I. Distribution, exposure, and human health risk analysis of heavy metals in drinking groundwater of Ghayen County, Iran

II. Protocol for the estimation of drinking water quality index (DWQI) in water resources: Artificial neural network (ANFIS) and Arc-Gis

III. Characteristics, water quality index and human health risk from nitrate and fluoride in Kakhk city and its rural areas, Iran

Thanks for highlighting these publication as they more informative, and we have adopted some of them to improve the introduction

Reviewer #2: Reviewer # comments:

Manuscript number: PONE-D-23-11494

Manuscript title: Burial leakage: a human accustomed groundwater contaminant sources and health hazards study near cemeteries in Benin City, Nigeria

Dear Authors, I have now read and assessed your manuscript with the abovementioned details. The topic of the manuscript is interesting. However, I have some concerns that should be carefully addressed. Please, see my queries/specific comments below.

Many thanks to your comments, they are very invaluable.

1. Please sufficiently justify that the selected parameters would provide signatures of leachates that are peculiar to cemeteries. Check whether it is really essential to emphasize on the impact of cemetery.

We have now sufficiently justified the statement in the revised manuscript.

2. Please emphasize more on the gaps in knowledge and novelty of the present study in the introduction section.

We have emphasized more on the gap in knowledge and novelty of our study.

3. This statement is unclear: “Several studies affirmed that this formation which follows the Oligocene-Pleistocene era in the continent of Africa at the sub-oceanic is underlain by sedimentary formation of the South Sedimentary Basin.” Please refine to make geologic sense.

We have reworded the statement to make more meaning than before.

4. “Benin formation” should be written as “Benin Formation”. The norm is that the word “formation” should be capitalized.

This has been formalised.

5. Please provide sufficient details on the surface hydrology, topography, and hydrogeology. These are very important for a better understanding of the influence of cemetery on the water quality.

We have now provided more details on the surface hydrology, topography, and hydrogeology of our study area. 

6. Geologic/hydrogeologic map is missing.

This was not included because it was not among the objectives of our study as “Omorogieva OM, Imasuen O. Litho-stratigraphic and hydrogeological evaluation of groundwater system in parts of Benin Metropolis, Benin City Nigeria: The key to groundwater sustainability. J. Appl. Sci. Environ. Manage. 2018; 22: 275 – 280” have provided the required details in their study. 

7. The results and discussion section should be robust. Compare your findings with literature data from other regions.

Attended to as suggested.

8. Please refer to these recent articles to improve the literature review, methodology and discussions: https://doi.org/10.1007/s13201-020-01180-9;
https://doi.org/10.1080/15275922.2021.2006363;
https://doi.org/10.1007/s10661-022-09789-w;
https://doi.org/10.1080/10106049.2022.2034990;
https://doi.org/10.1016/j.heliyon.2023.e15483;
https://doi.org/10.1016/j.ecolind.2023.110287;
https://doi.org/10.1007/s11356-023-26396-5

Thanks for highlighting these publication as they are more informative, and we have adopted some of them to improve literature review, methodology and discussions

9. The quality of the figures is poor. Please replace them with better versions.

We have now provided improved TIFF image. Please note that the journal may not provide reviewers with high quality image during the revision stage.

10. The novelty and significances of the current study should also be justified in the conclusions.

This has been done.

11. What are the limitations of the present study? Highlight on them.

We have added limitations of the research study to the manuscript

12. Also, discuss the perspectives for future research.

This has been included to the manuscript.

Thanks.

---

## [Decision Letter · Decision Letter 1]

12 Sep 2023

Burial leakage: a human accustomed groundwater contaminant sources and health hazards study near cemeteries in Benin City, Nigeria

PONE-D-23-11494R1

Dear Dr. Uyi,

We’re pleased to inform you that your manuscript has been judged scientifically suitable for publication and will be formally accepted for publication once it meets all outstanding technical requirements.

Kind regards,

Venkatramanan Senapathi, Ph.D.

Academic Editor

PLOS ONE

Additional Editor Comments (optional):

Reviewers' comments:

Reviewer's Responses to Questions

**Comments to the Author**

1. If the authors have adequately addressed your comments raised in a previous round of review and you feel that this manuscript is now acceptable for publication, you may indicate that here to bypass the “Comments to the Author” section, enter your conflict of interest statement in the “Confidential to Editor” section, and submit your "Accept" recommendation.

Reviewer #1: All comments have been addressed

Reviewer #2: All comments have been addressed

2. Is the manuscript technically sound, and do the data support the conclusions?

Reviewer #1: Yes

Reviewer #2: Yes

3. Has the statistical analysis been performed appropriately and rigorously? 

Reviewer #1: Yes

Reviewer #2: Yes

4. Have the authors made all data underlying the findings in their manuscript fully available?

Reviewer #1: Yes

Reviewer #2: Yes

5. Is the manuscript presented in an intelligible fashion and written in standard English?

Reviewer #1: Yes

Reviewer #2: Yes

6. Review Comments to the Author

Reviewer #1: The topic of this manuscript is good and written in good manner. All my comments are adrresed.

Therfore, I recommend it's publish.

Reviewer #2: Reviewer # comments:

Manuscript number: PONE-D-23-11494R1

Manuscript title: Burial leakage: a human accustomed groundwater contaminant sources and health hazards study near cemeteries in Benin City, Nigeria

Accept.

Best wishes.

7. PLOS authors have the option to publish the peer review history of their article (what does this mean?). If published, this will include your full peer review and any attached files.

Reviewer #1: **Yes: **Ahmad Zarei

Reviewer #2: No

---

## [Editor Report · Acceptance letter]

19 Sep 2023

PONE-D-23-11494R1 

Burial leakage: a human accustomed groundwater contaminant sources and health hazards study near cemeteries in Benin City, Nigeria 

Dear Dr. Uyi:

I'm pleased to inform you that your manuscript has been deemed suitable for publication in PLOS ONE. Congratulations! Your manuscript is now with our production department. 

Kind regards, 

on behalf of

Dr. Venkatramanan Senapathi 

Academic Editor

PLOS ONE